# New Apatite Fission-Track Data from the Murmansk Craton, NE Fennoscandia: An Echo of Hidden Thermotectonic Events

**Roman V. Veselovskiy [1,2,\*], Róbert Arató [3], Tanya E. Bagdasaryan [1,2], Alexander V. Samsonov [4], Alexandra V. Stepanova [5], Andrey A. Arzamastsev [6,7] and Mariya S. Myshenkova [1]**

[1] Geological Faculty, Lomonosov Moscow State University, 119991 Moscow, Russia; tanya.bagdasaryan@yandex.ru (T.E.B.); zf12@rambler.ru (M.S.M.)

[2] Institute of Physics of the Earth, Russian Academy of Sciences, 123242 Moscow, Russia

[3] Isotope Climatology and Environmental Research Centre, Institute for Nuclear Research, H-4001 Debrecen, Hungary; arato.robi@gmail.com

[4] Institute of Geology of Ore Deposits Petrography Mineralogy and Geochemistry, Russian Academy of Sciences, 119017 Moscow, Russia; samsonovigem@mail.ru

[5] Institute of Geology, Karelian Research Centre, Russian Academy of Sciences, 185910 Petrozavodsk, Russia; sa07sa@mail.ru

[6] Institute of Precambrian Geology and Geochronology, Russian Academy of Sciences, 199034 Saint Petersburg, Russia; arz1998@yahoo.com

[7] Institute of Earth Sciences, Saint Petersburg State University, 199034 Saint Petersburg, Russia

[\*] Correspondence: roman.veselovskiy@ya.ru

**Abstract:** For a long time, the thermal history of northeastern (NE) Fennoscandia in the Phanerozoic and Precambrian remained unknown, since no thermochronological studies were carried out within the Kola Peninsula area. Two years ago, we developed the first model of tectono-thermal evolution of the Kola Peninsula territory for the last 1.9 Gyr using a set of newly obtained apatite fission-track (AFT) and Ar/Ar thermochronological data. However, the low-temperature history of the most ancient tectonic unit of the northeastern part of the Kola Peninsula—the Archean Murmansk craton—remained poorly constrained due to the lack of AFT data. In this paper, we present the first results of AFT studies of 14 samples representing intrusive and metamorphic Precambrian rocks, located within the Murmansk craton of NE Fennoscandia. AFT ages and track length distributions indicate a similar tectono-thermal evolution of Precambrian tectonic units in NE Fennoscandia over the last 300 Myr. The AFT ages are distributed between ca. 177 and ca. 384 Ma; their median value, ~293 Ma, confirms the presence of a previously identified hidden thermal event that took place at about 300 Ma. However, a detailed analysis of the AFT age distribution shows the presence of three statistically distinguishable age components: 180–190 Ma (C1), 290–320 Ma (C2) and 422 Ma (C3). We assume that the relatively young AFT ages of C1 may originate from apatite crystals with low thermal resistivity. Remarkably, this value coincides with the initial stage of the Barents Sea magmatic province activity during large-scale plume-lithospheric interaction, as well as with the assumed age of an enigmatic remagnetization event throughout the Kola Peninsula. C2 ages can be observed in both the gabbroic and non-gabbroic samples, whereas C3 ages can only be found in gabbro. It is supposed that C2 ages, similarly to the Central Kola terrane, correspond to a cooling event related to the denudation of a thick sedimentary cover, representing a continuation of the Caledonian foreland basin towards NE Fennoscandia. C3 ages may be associated with a thermal event corresponding to the Caledonian collisional orogeny.

**Keywords:** apatite fission-track dating; Fennoscandia; Kola Peninsula; thermal history; tectonics; Phanerozoic; remagnetization

## 1. Introduction

The need for thermal history reconstruction in NE Fennoscandia within the Kola Peninsula area was formulated for the first time in [1]. It was expected to provide inevitable information for understanding the key tectono-thermal events, including the potential source of remagnetization in the Kola Peninsula Devonian dykes with a suggested age of 190 Ma [2–4]. However, up to now there have been no known thermal or magmatic events of this age within the Kola Peninsula area in general and the Murmansk craton in particular.

Fission-track (FT) analysis is a sensitive low-temperature thermochronological method that is widely used to construct quantitative models of the thermal and tectonic evolution of various tectonic units [5]. Until 2015, there were only two available apatite fission-track (AFT) ages for the territory of the Kola Peninsula [5], originally published in a PhD thesis [6], and they are virtually unavailable for assessing their quality and reliability. Thus, systematic low-temperature thermochronological studies were urgently needed in order to develop a quantitative model of the tectono-thermal evolution of northeast Fennoscandia.

The first modern AFT studies within the Kola Peninsula area were performed on drill core samples from different depths at the Khibina massif [1] (Figure 1), hosted in the Precambrian rocks of the Central Kola terrane. A set of 11 AFT ages in the range of 290-268 Ma had been obtained there, which correspond to the time elapsed roughly since the cooling of the massif below 110 °C. Later, we used 12 new AFT ages and 26 Ar/Ar ages from Phanerozoic and Precambrian magmatic bodies within the Central Kola terrane (Figure 1) for developing the first quantitative model of tectono-thermal evolution in NE Fennoscandia during the last 1.9 Gyr [7]. It was concluded that at about 300 Ma, the present-day surface of the central and southern parts of the Kola Peninsula cooled below 110 °C; we associated this cooling stage with the erosion of the sedimentary cover and/or with the drift of NE Fennoscandia over the large low-shear-velocity province in the deepest mantle. However, due to the absence of AFT data in the northernmost part of NE Fennoscandia within the tectonically distinct Archean Murmansk craton, this tectono-thermal model remains incomplete. The heterogeneity of the basement of the Murmansk craton and the Central Kola terrane, including the differences in their petrophysical properties, could potentially lead to their differing post-Devonian thermal evolution. Therefore, the low-temperature thermal history of the Murmansk craton obviously needs to be understood. In this article, we present new AFT data obtained from 14 Precambrian samples located within the Murmansk craton and their interpretation in a regional thermotectonic framework.

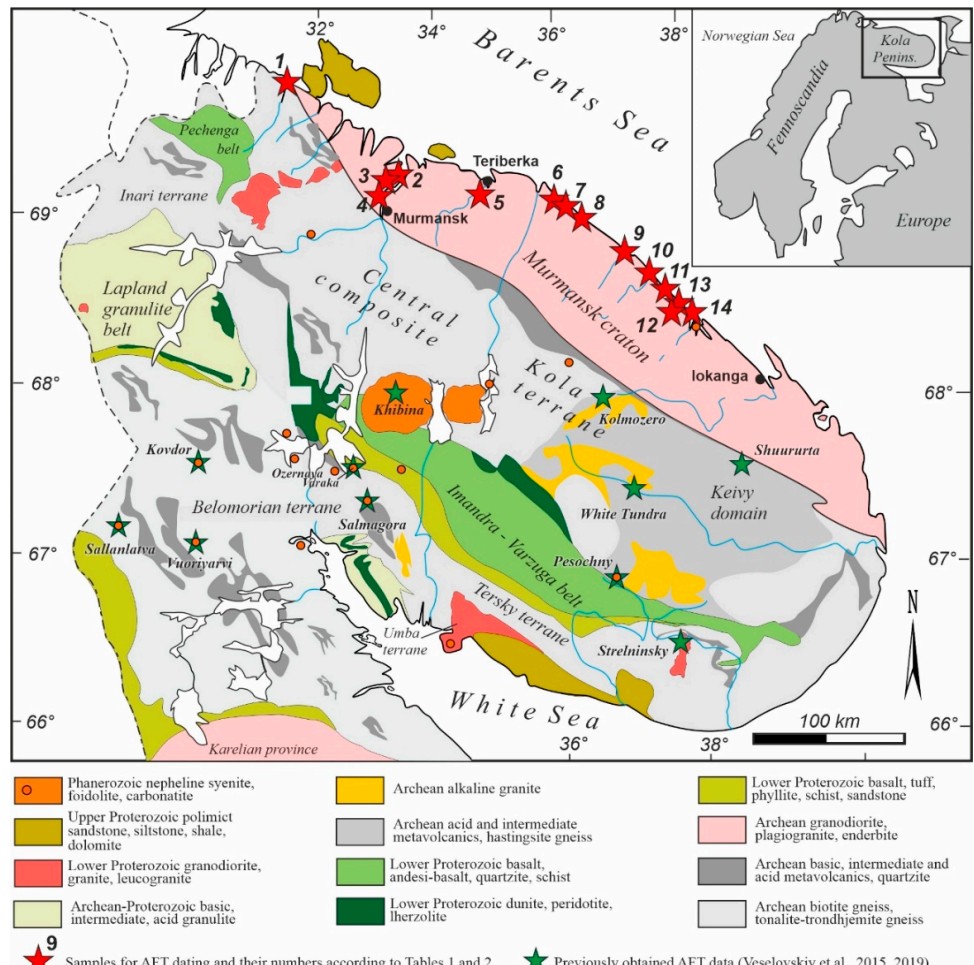

**Figure 1.** Schematic tectonic map of the Kola Peninsula showing sampling localities. AFT—apatite fission-track.

## 2. Geologic Setting and Sampling

### 2.1. Geological Background

The Murmansk craton represents Archean crust traced along the Barents Sea coast for ~600 km from the Sredniy Peninsula to the east (Figure 1). It consists of tonalite–trondhjemite–granodiorite (TTG) gneisses with sparse inclusions of mafic rocks, that are considered as the remnants of reworked greenstone belts [8]. The Murmansk craton is separated from the adjacent Central Kola terrane by a series of deep-seated faults [9]. The geochronological data for Teriberka enderbites (2772 ± 2 Ma, zircon, U-Pb SIMS) and Iokanga amphibolites, gneisses, enderbites, diorites, trondhjemites and granites (2717–2790 Ma, zircon, U-Pb SIMS) suggest that the main crust-forming events in the Murmansk craton occurred at 2.7–2.8 Ga [8]. These rocks' $T^{Nd}_{DM}$ values vary from 2.73 to 3.03 Ga [8,10], suggesting a relatively short crustal residence time for the source of granitoids. At least five episodes of mafic magmatism at 2.68, 2.50, 1.98, 1.86 and 0.38 Ga are also distinguished in the Murmansk craton according to U-Pb baddeleyite and zircon dating results [11–14].

### 2.2. Sampling

Apatite for fission-track analysis was separated from rocks representing the Precambrian magmatic and metamorphic complexes (Table 1), widely spread over the southern Barents Sea margin of the Kola Peninsula within the Murmansk craton (Figure 1, red stars). All 14 studied samples were taken from outcrops at altitudes not exceeding 100–150 m above sea level.

**Table 1.** Samples used for apatite fission-track analysis.

| NN | Sample No. | Latitude, °N * | Longitude, °E * | Locality | Rock name | Age, Ma | Method | Reference |
|----|-----------|---------------|----------------|----------|-----------|---------|--------|-----------|
| 1 | 514-1 | 69.689320 | 31.379680 | Liinakhamari | Tonalitic gneiss | 2780 | U-Pb zircon (SIMS) | [15] |
| 2 | 532-4 | 69.221293 | 33.418255 | Snezhnogorsk | Tonalitic gneiss | 2789 | U-Pb zircon (SIMS) | [15] |
| 3 | 533-4 | 69.222046 | 33.416460 | Snezhnogorsk | Gabbro pegmatite | 2500 | U-Pb baddeleyite (ID-TIMS) | [14] |
| 4 | 539-5 | 69.112630 | 33.383870 | Retinskoe | Gabbro pegmatite | 1983 | U-Pb baddeleyite (ID-TIMS) | [16] |
| 5 | 548-1 | 69.058179 | 35.008604 | Teriberka | Granodiorite | 2730 | The age is estimated by indirect evidence | A.V. Samsonov, unpubl. |
| 6 | 552-15 | 69.120363 | 36.054145 | Dalnie Zelentsy | Granite | 2730 | The age is estimated by indirect evidence | A.V. Samsonov, unpubl. |
| 7 | 563-3 | 69.115178 | 36.108886 | Dalnie Zelentsy | Tonalitic gneiss | 2836 | U-Pb zircon (SIMS) | [15] |
| 8 | 682-1 | 69.011853 | 36.531298 | Tryaschina Bay | Tonalitic gneiss | 2845 | U-Pb zircon (SIMS) | [15] |
| 9 | 680-5 | 68.831032 | 37.181274 | Chegodaevka Mt. | Tonalitic gneiss | 2850 | U-Pb zircon (SIMS) | [15] |
| 10 | 631-5 | 68.646679 | 37.771174 | Cape Litsky | Tonalitic gneiss | 2827 | U-Pb zircon (SIMS) | [15] |
| 11 | 639-3 | 68.591123 | 37.880255 | Mertvetskaya Bay | Granite | 2822 | U-Pb zircon (SIMS) | [15] |
| 12 | 676-4 | 68.440881 | 38.211200 | Dvorovaya Bay | Gabbro pegmatite | 1860 | U-Pb baddeleyite (ID-TIMS) | [17] |
| 13 | 676-3 | 68.441035 | 38.211450 | Dvorovaya Bay | Gabbro pegmatite | 1860 | U-Pb baddeleyite (ID-TIMS) | [17] |
| 14 | 675-3 | 68.378845 | 38.379579 | Varzina Bay | Granite | 2745 | U-Pb zircon (SIMS) | [15] |

* Map datum WGS84.

## 3. Methods

### 3.1. Apatite Fission-Track (AFT) Dating

Apatite grains of 250–50 μm size were concentrated by conventional heavy mineral separation techniques including crushing and sieving as well as magnetic- and heavy-liquid separation. Handpicked grains were mounted in epoxy resin then ground and polished in two (P1200 and P2500) and three (9, 3 and 1 μm) steps, respectively. Spontaneous fission tracks were revealed by etching with 5 M $HNO_3$ at 20 °C for 20 s. Samples were analyzed by the external detector method (EDM) [18] using very low uranium muscovite mica detectors, and irradiated at the Oregon State University Triga Reactor, Corvallis, USA. Neutron fluence was monitored using CN5 uranium-doped glass. After irradiation, induced tracks in the mica external detectors were revealed by etching with 40% HF for 20 min. Fission tracks were counted and confined track lengths were measured by R.A. at the Institute for Nuclear Research of Hungary by an Olympus BX53 microscope equipped with a microscope-computer-controlled stage system [19]. Reported confined track lengths include "track in track" (~80–90% of the confined tracks) and "track in cleavage" measurements as well (Table S1). Central ages [20,21] were calculated by the TRACKKEY software (version 4.2.g) [22] based on the approach of [23]. A personal zeta-calibration factor of 289.8 ± 4.9 was obtained by calibration against the Durango apatite and Fish Canyon Tuff apatite age standards according to the recommendations of [24].

### 3.2. Thermal History Modeling

In order to constrain the thermal history of the studied samples consistent with apatite fission-track ages and track length distributions, we used HeFTy software version 1.8.6 [25]. HeFTy is a thermal history modeling software, in which formalized statistical hypothesis tests assess the goodness-of-fit between the input AFT data and AFT data predicted from many thousands of time-temperature (t-T) paths according to the [26] AFT annealing model [27]. The output of the model is two t-T envelopes: the broader envelope being the range within which any thermal history cannot be excluded by the measured data (acceptable fit); the narrower envelope being the range that is supported by the measured data (good fit). The merit values used are "0.05" for acceptable fit and "0.50" for good fit. For all cases, simulations were run until 100 good-fit paths were found. We applied the present-day annual surface temperature for the Kola Peninsula of $T_{ann} = 0$ °C (Met Office: www.metoffice.gov.uk/).

## 4. Results

Fourteen samples yielded AFT central ages ranging from ca. 384 to ca. 177 Ma (1σ uncertainty), with a mean of 292.2 ± 48.3 Ma (Table 2, Figure 2). In general, the obtained AFT ages of most samples are statistically indistinguishable within 1σ uncertainty, with an exception of sample #11 (639-3) as discussed below. It is worth noting that the gabbros (samples ## 3, 4, 12, 13) show the oldest ages in the obtained dataset.

Table 2. Apatite fission-track data.

| NN | Sample No. | No. of Crystals | Track Density (×10⁶ tracks/cm²) (No. of Tracks) | | | Mean Dpar (μm) | Mean U (ppm) | Age Dispersion (Pχ²) | Central Age (Ma) (±1σ) | Apatite Mean Track Length (μm ± 1 s.e.) (No. of Tracks) | Standard Deviation (μm) |
|---|---|---|---|---|---|---|---|---|---|---|---|
| | | | $P_s$ ($N_s$) | $P_i$ ($N_i$) | $P_d$ ($N_d$) | | | | | | |
| 1 | 514-1 | 12 | 11.947 (233) | 9.332 (182) | 15.13 (31,666) | 2.81 | 7.95 | 0.00 (96.53%) | 274.7 ± 27.6 | 13.78 ± 0.5 (5) | 1.04 |
| 2 | 532-4 | 24 | 24.691 (511) | 17.588 (364) | 16.02 (31,666) | 2.50 | 12.82 | 0.00 (99.44%) | 317.9 ± 22.5 | 13.53 ± 0.3 (39) | 1.84 |
| 3 | 533-4 | 4 | 13.4 (72) | 9.492 (51) | 16.25 (31,666) | 2.63 | 6.89 | 0.00 (79.0%) | 324.2 ± 59.6 | 13.65 ± 0.5 (2) | 0.49 |
| 4 | 539-5 | 23 | 7.162 (310) | 5.337 (231) | 16.91 (31,666) | 3.26 | 3.69 | 0.00 (96.08) | 320.7 ± 28.5 | 14.40 ± 0.4 (7) | 1.08 |
| 5 * | 548-1 | 25 | 34.742 (840) | 32.674 (790) | 17.14 (31,666) | 2.30 | 24.2 | 0.00 (98.87%) | 258.8 ± 13.6 | 13.80 ± 0.2 (50) | 1.36 |
| 6 * | 552-15 | 25 | 23.533 (1145) | 17.059 (830) | 13.8 (31,666) | 1.82 | 14.66 | 0.01 (84.25%) | 270.1 ± 13.2 | 13.31 ± 0.1 (100) | 1.00 |
| 7 | 563-3 | 25 | 8.493 (469) | 5.686 (314) | 14.02 (31,666) | 1.87 | 4.83 | 0.01 (78.84%) | 296.5 ± 22.3 | 13.71 ± 0.3 (27) | 1.41 |
| 8 * | 682-1 | 26 | 21.853 (985) | 15.663 (706) | 14.69 (31,666) | 2.17 | 13.05 | 0.00 (98.73%) | 290.4 ± 15.2 | 13.61 ± 0.1 (100) | 1.18 |
| 9 | 680-5 | 25 | 10.393 (667) | 7.495 (481) | 14.24 (31,666) | 2.36 | 6.42 | 0.00 (92.4%) | 280.0 ± 17.5 | 12.98 ± 0.3 (21) | 1.65 |
| 10 | 631-5 | 25 | 13.716 (767) | 9.782 (547) | 14.91 (31,666) | 2.39 | 8.87 | 0.01 (71.76%) | 296.1 ± 17.4 | 13.20 ± 0.2 (20) | 1.08 |
| 11 * | 639-3 | 23 | 31.119 (1028) | 41.351 (1366) | 16.47 (31,666) | 2.59 | 31.8 | 0.00 (92.41%) | 177.2 ± 8.0 | 13.08 ± 0.2 (74) | 1.50 |
| 12 | 676-4 | 24 | 17.783 (890) | 10.53 (527) | 14.47 (31,666) | 2.06 | 8.65 | 0.05 (48.11%) | 344.7 ± 19.9 | 13.59 ± 0.2 (45) | 1.35 |
| 13 * | 676-3 | 57 | 10.641 (1261) | 6.076 (720) | 15.58 (31,666) | 2.50 | 4.67 | 0.26 (0.07%) | 383.8 ± 19.2 | 13.27 ± 0.2 (103) | 1.58 |
| 14 | 675-3 | 21 | 43.216 (817) | 38.032 (719) | 15.8 (31,666) | 2.71 | 35.03 | 0.00 (89.62%) | 255.1 ± 13.8 | 13.43 ± 0.3 (24) | 1.34 |

Notes: (i). Analyses by external detector method; (ii). Ages calculated using dosimeter glass: CN5 and ζ = 289.8 ± 4.9 (apatite); (iii). Pχ² is the probability of obtaining a χ² value for v degrees of freedom where v = no. of crystals—1; (iv). s.e. = standard error of the mean; (v). U ppm calculated from induced track density relative to CN5 glass. *—Apatite fission-track (AFT) data, that meet the modern reliability criteria [28]: number of crystals counted ≥ 20, number of track lengths measured ≥ 50.

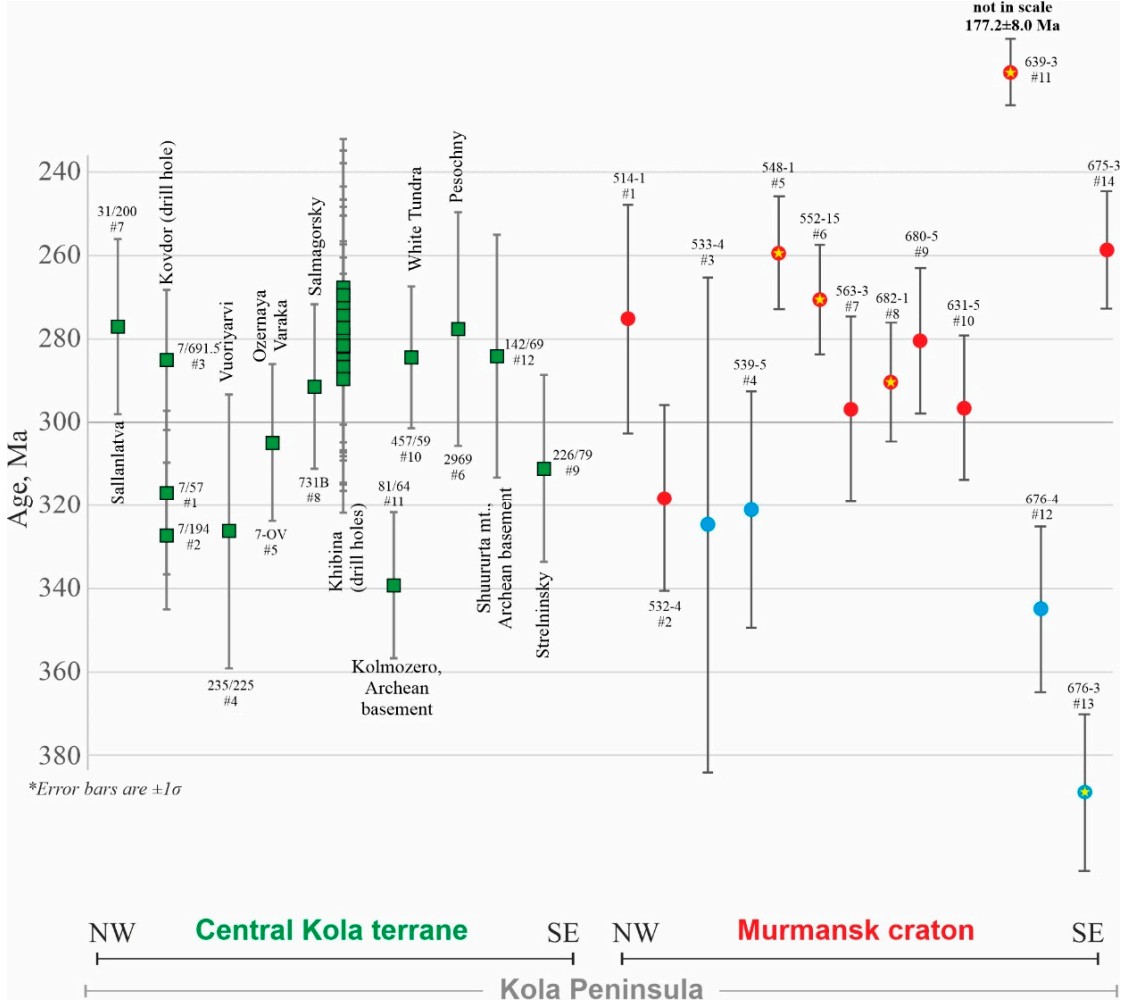

**Figure 2.** Distribution of the Kola Peninsula apatite fission-track ages: green—previously published [1,7], red and blue—this work. Blue points—AFT ages of gabbro samples (see text). Sample numbers are given according to the corresponding tables in cited works and Tables 1 and 2 (Murmansk craton, this study). Points with yellow stars mark the AFT data, that meet the modern reliability criteria [28]: number of crystals counted ≥20, number of track lengths measured ≥50. For each tectonic unit (Central Kola terrane and Murmansk craton), samples are arranged from NW to SE.

Mean track lengths vary between 14.40 ± 0.4 and 13.08 ± 0.2 µm; the most reliable ($n \geq 50$) mean track length distributions were obtained for five samples (Table 2). Track length distributions for all samples are unimodal (Figure 3), which indicates moderate to fast monotonic cooling without periods of long residence within the partial annealing zone.

Complete AFT data including individual grain age data, track length data and $D_{par}$ values are presented in Supplementary Materials, Table S1.

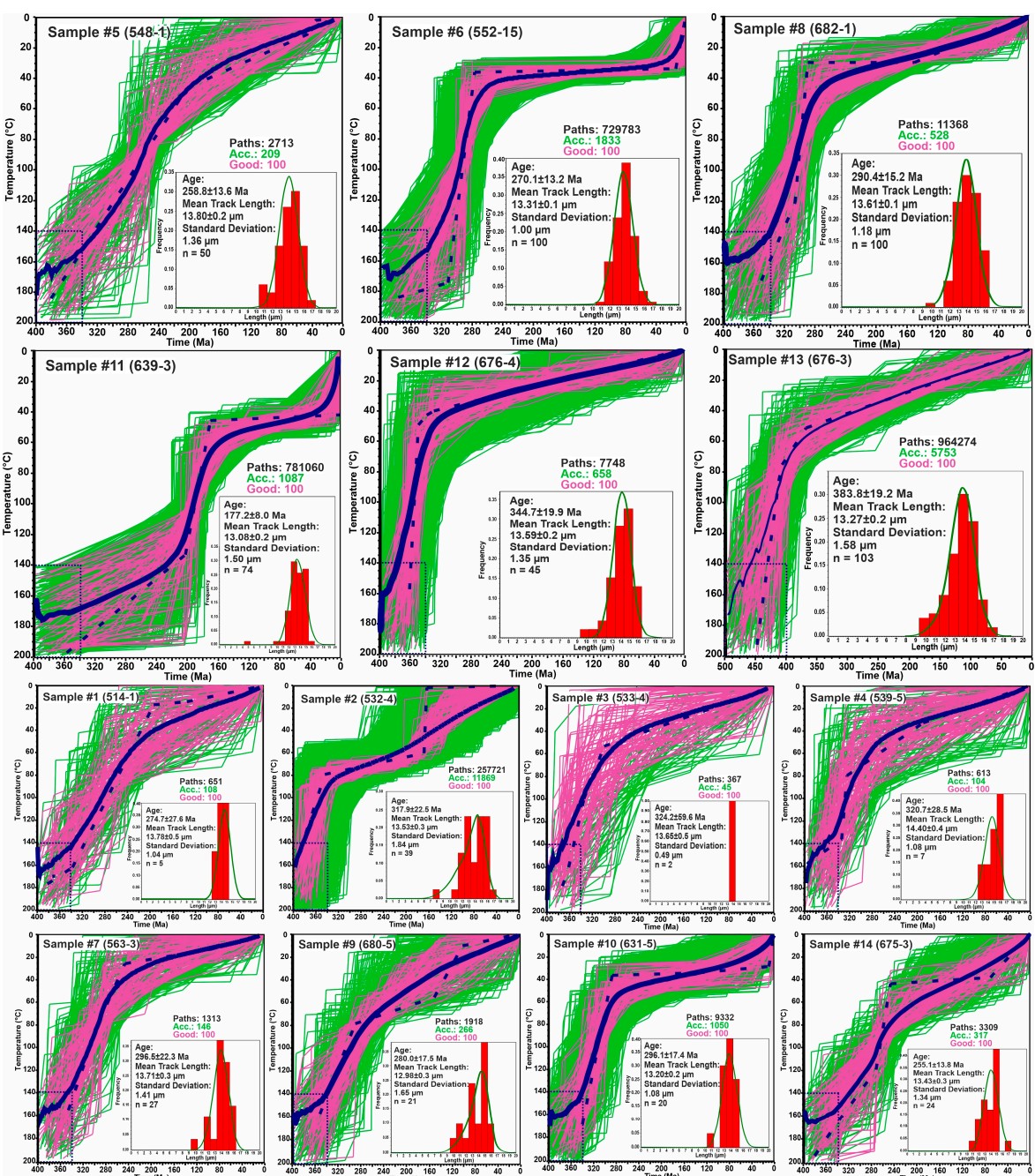

**Figure 3.** Results of the thermal modeling showing time-temperature (t-T) histories that explain the measured apatite fission-track (AFT) age and track length data. Results are shown as path envelopes encompassing the thermal histories that generated good fits (purple curves) and acceptable fits (green curves) to the data. The best fit path is shown in dotted blue line and the weighted mean path in continuous blue line. Insets: the measured track length distribution histograms (red) and predicted length distributions (green lines). Dashed boxes are constraint boxes. Note that t-T histories for sample #13 (676-3) begin at 500 Ma. Small plots indicate less reliable thermal histories due to the low number of available confined track lengths in those samples

## 5. Discussion

### 5.1. Time-Temperature Modeling

AFT data convincingly demonstrate that all studied Archean–Proterozoic dykes and metamorphic complexes of the basement of the Murmansk craton (NE Fennoscandia) were completely reset around

300 Ma. The obtained results are in an agreement with the previous ones from the Kola Peninsula (Figure 2) and indicate a similar post-Devonian tectono-thermal history of the Central Kola terrane and Murmansk craton. This conclusion is supported by the t-T models of the studied samples (Figure 3), which indicate that, apart from a few exceptions that will be discussed below, the thermal evolution of the Murmansk craton is in general agreement with the previously proposed models for the thermal evolution of the Kola Peninsula. Despite the generally consistent modeling results, it has to be noted that only five samples yielded sufficient AFT ages and confined track length data for a well constrained thermal history model (large plots in Figure 3 except for the sample #12). Therefore, several samples resulted in less constrained thermal models (smaller plots in Figure 3 including the sample #12) where more detailed future work with a significantly larger amount of confined track length measurements would be desirable.

Mean track length (MTL) distributions (Figure 3) and AFT ages show that the rocks of the present-day erosional surface cooled below ca. 110 °C about 300 Ma; several tens of millions of years later they left the apatite partial annealing zone (APAZ). The cooling rate of the craton at this stage can be estimated at approximately 1–2 °C/Myr. Subsequently, it slowly cooled to modern temperatures due to erosion at a rate of about 0.1–0.2 °C/Myr. This is in good agreement with the estimates of the exhumation rates obtained for the other cratons [29]. In a few cases, the best-fit t-T paths show a third cooling event in the interval 50–0 Ma (Figure 3), during which there was an increase in the rate of cooling down to modern temperatures with a cooling rate of ~0.4 °C/Myr. We believe that this stage is not tectonic in nature but is the result of a global decrease in the surface temperature in the Cenozoic [30,31]; this issue was discussed in detail in [1].

## 5.2. "Too Young" and "Too Old" AFT Ages

Among the obtained AFT ages there are some which differ significantly from the overall mean. Here, we will discuss the possible origin of "too young" and "too old" AFT ages of the studied rocks.

Figure 4a shows the dependence of the AFT age on the uranium concentration in the studied samples from the Murmansk craton. Two age groups are statistically significant: (1) the age group (age component C1—193 ± 13 Ma) obtained from apatite grains with relatively high uranium concentration, and (2) the second group of old ages (age component C2—318 ± 10 Ma) obtained from grains with a relatively low uranium content. If we exclude the data obtained for gabbro from the total selection of AFT ages (see below), then the age component values will become somewhat younger: C1—180 ± 12 Ma and C2—289.4 ± 9.3 Ma (Figure 4b).

Before discussing possible geological interpretations of "extreme" ages, we consider the likely factors causing distortions in the fission-track age of apatite. First, we considered the possibility of AFT age rejuvenation due to the high uranium content in apatite, which leads to the so-called radiation-enhanced annealing (REA) effect—i.e., decrease in the stability of the crystal lattice of apatite to the fission-track annealing process. This effect has been discussed in great detail in systematic thermochronology studies across Scandinavia (e.g., [32–35]) and other cratons (e.g., [36–38]). Hendriks and Redfield [32] pointed out that many AFT ages in Fennoscandia are younger than the apatite (U-Th)/He ages in the same samples, although this should be the other way around. They concluded that AFT data obtained for the western regions of Finland (500 Ma and older) could be significantly affected by REA as compared to samples with AFT ages <300 Ma. On the contrary, [33] emphasized the importance of chlorine content over REA, which is at least partly supported by the dataset presented in [36]. Barry Kohn and co-authors also pointed out that gabbros tend to yield the oldest AFT ages in Fennoscandia and that negative eU-AFT age correlation is reduced when LA-ICP-MS is used for determining apatite U-contents [36]. The latter is related to the fact (the so-called counting bias) that AFT ages determined by EDM are not independent from U-contents determined from induced track densities and the two values will always show a negative correlation (e.g., [39,40]). A recent input by [38] further studied the eU (REA)-AFT age relationship.

They concluded, based on a large dataset including trace element data, that REA has a significant effect on AFT ages in low-Cl samples with protracted thermal histories (>200–500 m.y.) at <100 °C.

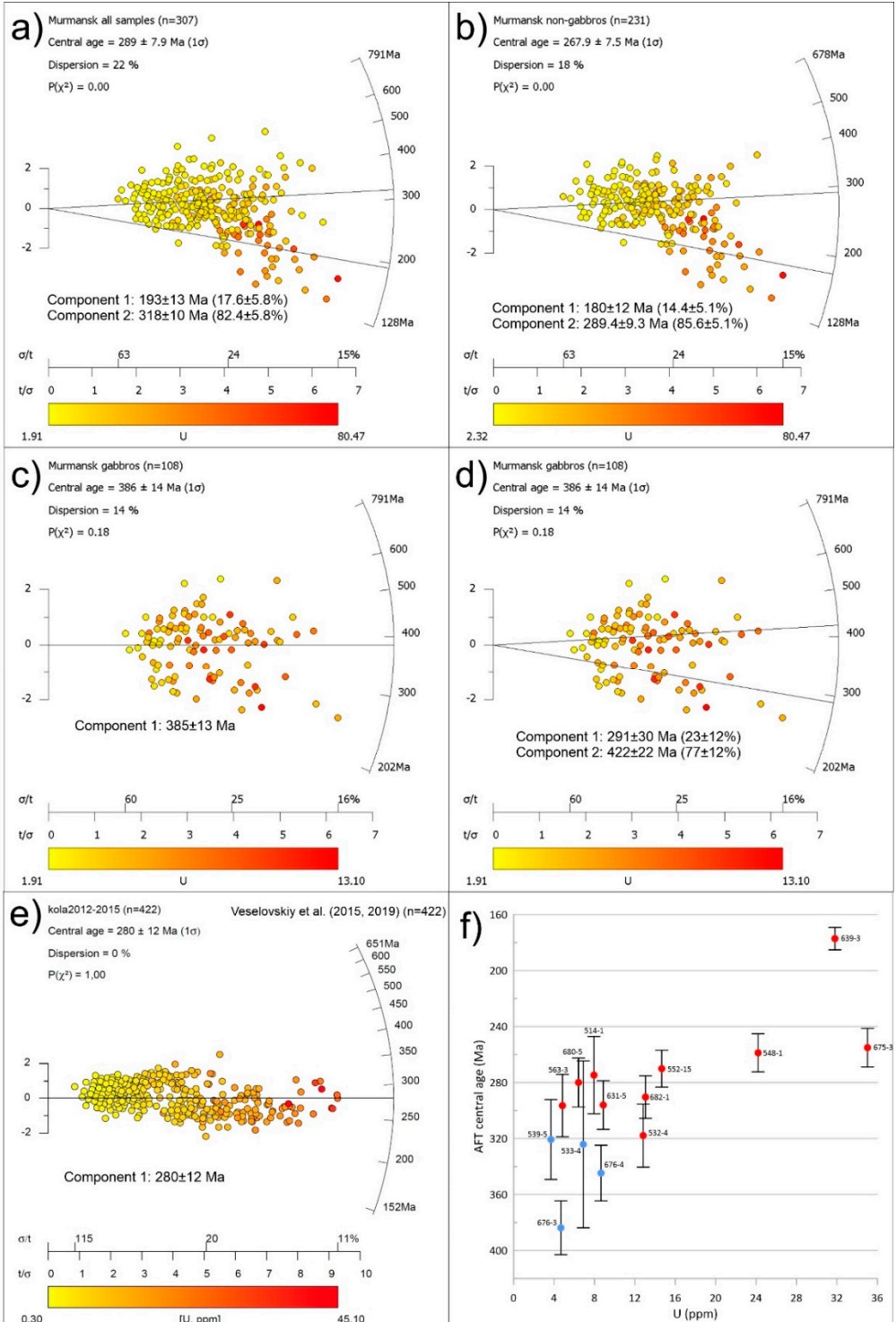

**Figure 4.** Radial plots presenting AFT age and uranium concentration dependence for (**a**) all samples studied in this work, (**b**) non-gabbro samples, (**c**,**d**) gabbro samples and (**e**) all previously studied samples from the Central Kola terrane [1,7]; (**f**) dependence of the mean U content in apatite from each sample with corresponding central AFT age; blue points represent the gabbro samples.

To test the possible effect of radiation-enhanced annealing on our AFT ages, we investigated their dependence on U-concentration determined by EDM (Table 2; Figure 4). Younger age components characterized by higher U-contents (Figure 4a,b) as well as the inverse correlation between AFT ages and U-content (Figure 4f) could possibly reflect REA. However, as mentioned above, AFT ages are not independent from induced track densities (Ni) due to the nature of the EDM (i.e., AFT ages are calculated using Ni). It is remarkable that the gabbroic samples, despite a large intra-sample variability of ages, do not show a U-concentration-related separation of ages (Figure 4c,d). We assume that this could be due to the lack of a spontaneous-track (Ns)-related counting bias—i.e., the fact that older ages can be determined only when the U-concentration is small enough to result in a countable amount of spontaneous tracks. This condition is given only in our gabbroic samples with a restricted range of generally low U-contents (and low U-contents in general). It should be noted that Ns-related counting bias is inherent to the nature of the fission-track method and independent on the approach used for determining U-concentrations (LA-ICP-MS or EDM). It can at least partly explain why there is a seemingly weaker/no eU-AFT age correlation in low-U and/or young samples (i.e., small Ns) in the current dataset and previously cited studies.

Although we cannot directly prove or disprove the effect of REA, Figure 4 implies that there are multiple apatite generations in our samples with largely different thermal histories or thermal resistivities. It is remarkable that the AFT age of 177 ± 8 Ma of granite sample #11 and the younger age component of our samples (ca. 190 Ma) corresponds to the time of a remagnetization event, proposed based on a large number of paleomagnetic data and selectively affected many Devonian dykes throughout the Kola Peninsula [2,3] and, in particular, in the Murmansk craton [4]. Among the possible sources of this remagnetization, we consider a hidden endogenous activity, associated with the initial stages of the Barents Sea Jurassic–Cretaceous magmatic province formation [41]. The fact that apatite from the neighboring studied objects has an AFT age of about 300 Ma may mean that the manifestations of this ~190 Ma endogenous activity were local and probably appeared within tectonically weakened (fluid-permeable) zones. A similar selectivity is typical for the aforementioned Mesozoic remagnetization event [4]. Alternatively, some of the dated grains (especially in sample #13 (676-3)) have a much lower thermal resistivity, which can record thermal events at lower temperatures than others.

We have evaluated the possible effect of radiation-enhanced annealing also on AFT ages in previously studied objects from the Central Kola terrane [1,7]. Figure 4e shows that there are no statistically significant age groups; also, there is no clear dependence between AFT ages and uranium concentration in the Central Kola terrane.

The AFT age of 388 ± 27 Ma in the gabbro pegmatite sample #13 (676-3) coincides with the manifestation of Devonian magmatism within the Kola Peninsula area [42]. This age could be expected based on regional geology, but such old AFT ages have not yet been known for the Kola Peninsula rocks. Our explanation for this result is as follows. If we have a closer look at the oldest samples (Figure 2, Murmansk craton), we notice that the oldest ages are related exclusively to gabbroic samples (Figure 2, blue points). Furthermore, by optically investigating the radial plots of these samples (Figure 4d), it seems that their single ages are roughly grouped into two populations around 291 and 422 Ma. However, these two groups are automatically distinguishable by the BinomFit (version 1.2) [43] or DensityPlotter (version 8.5) [44] software only if a larger number of single crystal ages are determined. This could only be achieved in the case of sample #13 (676-3), which also fails the so-called chi-square test (Table 2) suggesting that the individual ages of this sample do not derive from a single population [45]. In order to understand the mechanism behind these two apparent populations, we systematically measured the Dpar-s of the "young" and "old" populations, but there is seemingly no difference between them in this respect. We also measured the chlorine content of some of the "old" and "young" grains via Energy-Dispersive X-ray Spectroscopy (EDX) but found no systematic difference between the two apparent populations. As the "young" population is similar to other samples' central ages, it seems that the "old" population shows a much higher thermal retentivity

than the rest of the crystals. Given that most AFT ages are similar to those in other parts of the Kola Peninsula where several Ar/Ar ages below 400 Ma were reported [7], the ca. 422 Ma apparent population could represent an extremely retentive apatite variety and its age might reflect a thermal event related to the Caledonian orogenesis. However, the explanation of this phenomenon needs further investigation, including trace element (e.g., LA-ICP-MS) and structural (e.g., TEM) analysis.

### 5.3. New AFT Results in the Framework of Fennoscandia's AFT Map

Figure 5 represents a total coverage of Fennoscandia by AFT data as of 2007 [5] as well as the results of recent [1,7] and current AFT studies from the Kola Peninsula. It is obvious that the outer parts of Fennoscandia are characterized by younger AFT ages than its interiors, where the AFT ages can be as old as 700–800 Ma. Almost all AFT ages from the studied areas of the Kola Peninsula are ca. 300 Ma assuming a similar thermal evolution in this area. However, relatively young AFT ages (ca. 177 Ma) in the north (this study) and in the central part of the Kola Peninsula [5] suggest the presence of zones with potential thermal influence by the Barents Sea magmatic province. This explanation requires additional confirmation by future thermochronology studies within the inner parts of the Central Kola and Murmansk terranes. The western part of the Kola Peninsula (Russia/Finland border), as well as the vast territory in the south (the Belomorian terrane and the Karelian craton), remains completely unexplored by the AFT method.

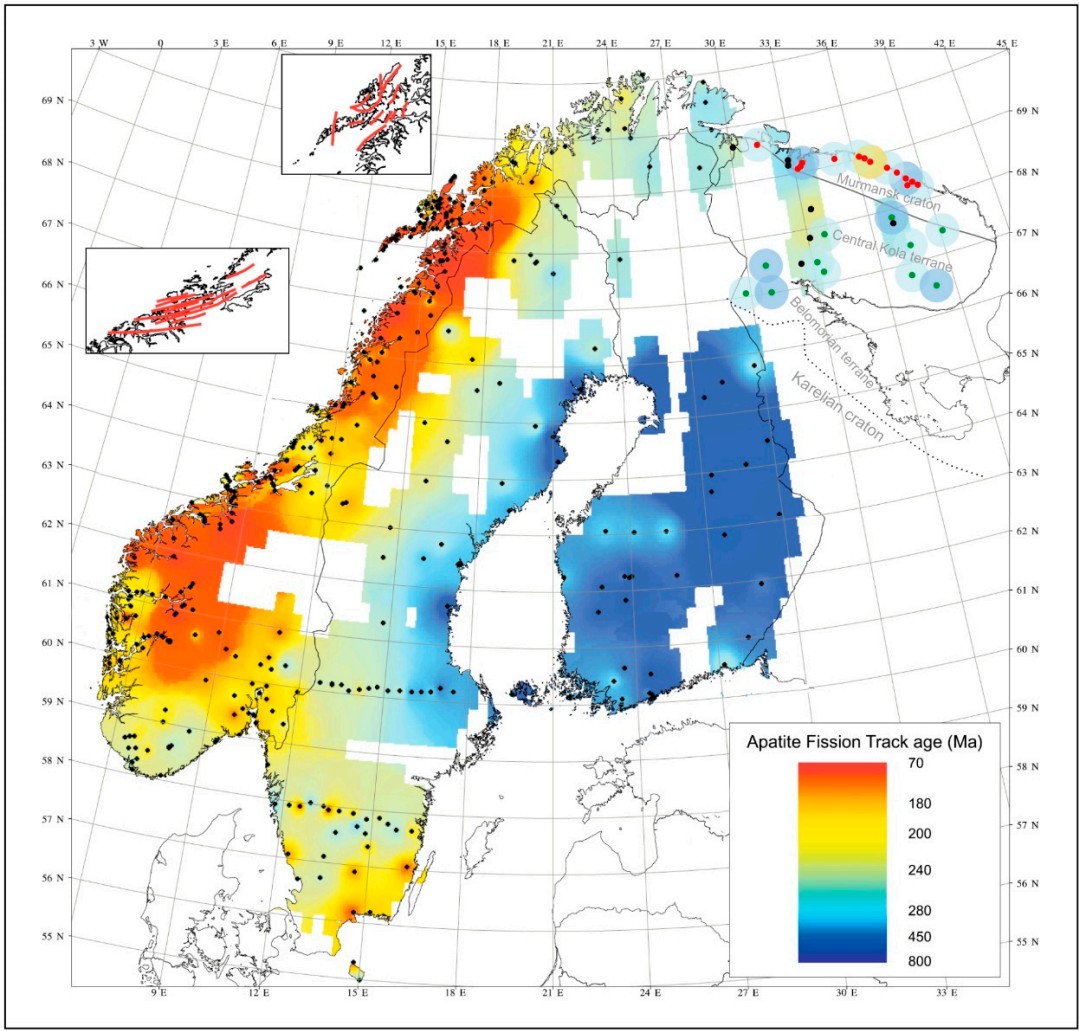

**Figure 5.** Apatite fission-track ages in Fennoscandia after [5] (black dots) with the Kola AFT data from [1,7] (green dots) and this study (red dots) added.

Thus, over the past 5 years we have made first but important steps towards understanding the Kola Peninsula's thermochronology. At the same time, several details of its thermal evolution remain largely unclear, and future AFT studies in the above-mentioned areas should help to decipher them.

## 6. Conclusions

The main results of the presented study are the following:

(1)　Apatite fission-track ages, representing the Precambrian intrusive and metamorphic complexes of the basement of the Murmansk craton of NE Fennoscandia, indicate that they were cooled below 110 °C at about 300 Ma; this cooling stage is associated with the general uplift of Fennoscandia and/or a decrease in heat flow [7];

(2)　time-temperature modeling indicates similar thermal evolution of the Murmansk craton and the Central Kola terrane in the Mesozoic and Cenozoic;

(3)　the youngest obtained AFT age (177 Ma) as well as the younger (ca. 190 Ma) age component detected in other samples is possibly associated with a hidden thermal event that led to the remagnetization of a number of dykes on the Kola Peninsula [4];

(4)　four relatively old AFT ages were obtained from gabbro samples, including the oldest one—385 Ma. A larger number of AFT single grain ages obtained from these samples reveal the presence of two age components—291 and 422 Ma. We assume that the 385 Ma central age has no geological meaning, but it is rather a mixture of the mentioned age components. It is possible that grains with AFT ages around 422 Ma, due to their chemical composition, are the most resistant to track annealing. In this case, the age component 422 Ma is associated with the active phase of Caledonian orogeny;

(5)　obtained results significantly increase the low-temperature thermochronological database of NE Fennoscandia and define the inner regions of the Kola Peninsula and the Karelian craton as the primary targets for future AFT studies.

**Supplementary Materials:** The following are available online at http://www.mdpi.com/2075-163X/10/12/1095/s1, Table S1: Raw AFT data.

**Author Contributions:** R.V.V., A.A.A., A.V.S. (Alexander V. Samsonov) and A.V.S. (Alexandra V. Stepanova) conceived and designed the study; they also conducted field works and some of the experiments; R.A., T.E.B. and M.S.M. conducted AFT analysis and t-T modeling. All authors discussed the results, problems and methods, and contributed to interpretation of the data and writing the paper. All authors have read and agreed to the published version of the manuscript.

**Funding:** Field work and mineral separation were supported by Russian Scientific Foundation, grant #16-17-10260. AFT studies were supported by the Ministry of Education and Science (grant MD-1116.2018.5) and by the RFBR (grant #20-35-90066). R.A.-s contribution to AFT dating was supported by the PPD2018-003/2018 Premium Postdoctoral Grant of the Hungarian Academy of Sciences as well as the European Union and the State of Hungary, co-financed by the European Regional Development Fund in the project of GINOP-2.3.2-15-2016-00009 "ICER".

**Acknowledgments:** We thank the editors and the anonymous reviewers for their constructive comments leading to substantial improvement of the manuscript.

**Conflicts of Interest:** The authors declare no conflict of interest.

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
