# Peer review of "New Apatite Fission-Track Data from the Murmansk Craton, NE Fennoscandia: An Echo of Hidden Thermotectonic Events"

_minerals, doi:10.3390/min10121095_

Round 1
Reviewer 1 Report
The structure of the manuscript , its content, research methods, graphics and references are made at a good level. In general, the manuscript meets the requirements for articles of the international level.
Author Response
Dear colleague,
thank you so much for high scoring of our manuscript!
Stay safe and healthy.
Best regards,
Roman Veselovskiy
Reviewer 2 Report
General introduction
This manuscript focuses on the tectono-thermal history of NE Fennoscandia. This study expands on the previous work by e.g. Hendriks et al. (2007) in an area with limited thermochronological data available. The tectono-thermal evolution is documented by thermal history modelling of apatite fission track data from fourteen new samples.
General comments
This manuscript provides an interesting story on the tectono-thermal evolution of the Murmansk Craton in NE Fennoscandia. The text is nicely written and easy to follow. The manuscript lacks a critical evaluation of own data, especially on the quality of the thermal history models. Most of the thermal history models are based on samples with so few track length measurements, so that one can practically model any thermal history. I would therefore strongly recommend the authors to conduct more track length measurements and re-model the samples. See comment for line 135.
Figures: The figures in this manuscript are nice and easily understandable. My comments on the figures are on Fig. 3:
- The size of the models should be similar in order to make it easier to compare.
- The resolution of the figure is too low. It is difficult to read the text (except sample number). My recommendation is to save the plot from HeFTy as PDF, and then import it into Adobe Illustrator (or similar software). Then you could create new axes and possibly also add chronostratigraphy on top.
- The figure needs a legend. The two blue lines are not explained so far.
Tables:
- There is a mismatch between sample NN and sample number in tables 1 and 2. These also need to be consistent with the figures. This is a problem when the samples are referred to in the text (e.g. line 242).
Supplementary material:
Including all the raw data is an excellent way for allowing other scientists to reproduce the thermal history modelling. There is, however, a mismatch between the number of reported ages in the supplementary material and in Table 2. There is also a mismatch between the number of counted grains and number of grains with Dpar measurements:
- 514-1: Ages have been reported for 12 grains in the supplementary material. 13 grains are reported in the table.
- 532-4: Dpars have been measured for 25 grains in the supplementary material, ages have been reported for 23 grains. 25 grains are reported in the table.
- 533-4: Ages have been reported for 4 grains in the supplementary material. 5 grains are reported in the table.
- 539-5: Dpars have been measured for 25 grains, ages have been reported for 23 grains in the supplementary material. 25 grains are reported in the table.
- 552-15: Ages have been reported for 25 grains in the supplementary material. 26 grains are reported in the table.
- 563-3: Ages have been reported for 25 grains in the supplementary material. 26 grains are reported in the table.
- 639-3: Dpars have been measured for 25 grains, ages have been reported for 23 grains in the supplementary material. 25 grains are reported in the table.
- 675-3: Ages have been reported for 21 grains in the supplementary material. 22 grains are reported in the table.
Line-specific comments
21 - 23: This sentence is misleading. The way this is written, it is understood as this paper also include Ar/Ar data. I would write something like: “Recently, apatite fission track (AFT) and Ar/Ar data have been used for developing…”.
29: Remove uncertainties. Ages can also be reported as c. 185 and c. 385 Ma. The proper ages are reported in the table later anyways.
38 – 39: This sentence is repetition.
49: This is a strange way to refer to a publication. The name of the first author should be included. I suggest to write something like: “…formulated for the first time by Veselovskiy et al. [1]”. Such referencing is also done later on in the manuscript.
62 – 78: This paragraph is somewhat confusing. When reading it, I get the impression that the work is included in the current paper, while it is from other publication by the same authors. I would suggest to write it in a passive language to avoid this confusion.
67: Maybe the amount of Ar/Ar data can be quantified?
84 – 85: Sentence starting with “Intrusive enderbites” is mostly repetition of the previous sentence. These should be merged.
113: It should be specified whether the measured confined tracks are TINTs (Track-IN-Tracks) or TINCLEs (Track-IN-CLEavage). According to the supplementary material, both types have been measured. According to (Donelick et al., 2005; and references therein), only TINTs should be measured. This can be because the width of the cleavages can be difficult to measure properly (and hence also the correct track length), but also because TINCLEs CAN be more resistant to annealing and possibly not anneal at all. In any case, the inclusion of TINCLEs can affect the final outcome of the thermal models.
116 and 118: Same as for line 49 (reference 20 and 21).
119 – 129: What constraint has been used as a starting point? Also, have any other thermal history scenarios been tested (e.g. exhumation and re-burial by sediments)?
124: Please check the style of referencing.
128: Maybe you can include the number of paths tested in the model? That would be a way to evaluate and compare the different models.
130: I suggest to highlight that the gabbros are the oldest as this is further discussed at lines 242 – 261.
131 – 132: Same comment as for line 29.
135: The number of confined track lengths in this study is mostly too low. According to Kohn et al. (2019), MTL values stabilize after 50 – 120 measurements. In this manuscript, only five out of fourteen samples have ≥ 50 TL measurements. This will strongly influence the modelling. By using a low number of TL measurements, one can basically model any thermal history one wants. My recommendation is therefore to conduct more TL measurements for the samples with too few measurements, maybe by making and etching additional mounts (no irradiation).
141: The table looks mostly nice. According to the supplementary material, only four grains have been counted for sample 3, not five. Also, you can consider to remove this entire sample as the uncertainties are way too high (few crystals counted; few TL measured). According to Kohn et al. (2019), typically more than 20 grains should be counted.
158 – 159: Sentence is difficult to understand. Please rewrite and simplify.
162: Uplift is defined as vertical displacement (of surface or rock) with respect to the Earth’s geoid, while exhumation is defined as the displacement of rocks with respect to the surface (England and Molnar, 1990). The thermal history models will record exhumation (either tectonic or due to erosion), not uplift.
182-184: Sentence is difficult to understand. Needs rewriting.
236 (Fig. 4C and D): The number of analyses in Fig. 4C and 4D (n = 108) does not match the total number of analyses for the gabbros reported in Table 2 (n = 111).
242: There is a mismatch in the tables for this sample. This sample also fails the chi-square test, indicating that it is likely that the age represents a mixed age.
282 (Fig. 5): This figure is very good for putting the new data into a larger context. However, at this stage, it is only briefly referred to in the very last sentence of the manuscript. I that this figure is either given more attention or removed from the manuscript.
REFERENCES
Donelick, R.A., O’Sullivan, P.B., Ketcham, R.A., 2005. Apatite fission-track analysis. Rev. Mineral. Geochemistry 58, 49–94. https://doi.org/10.2138/rmg.2005.58.3
England, P., Molnar, P., 1990. Surface uplift, uplift of rocks, and exhumation of rocks. Geology 18, 1173–1177. https://doi.org/10.1130/0091-7613(1990)018<1173:SUUORA>2.3.CO;2
Hendriks, B.W.H., Andriessen, P.A.M., Huigen, Y.D., Leighton, C., Redfield, T.F., Murrell, G.R., Gallagher, K., Nielsen, S.B., 2007. A fission track data compilation for Fennoscandia. Nor. Geol. Tidsskr. 87, 143–155.
Kohn, B., Chung, L., Gleadow, A., 2019. Fission-track analysis: field collection, sample preparation and data aquisition, in: Fission-Track Thermochronology and Its Application to Geology. pp. 25–48.

Author Response
Dear colleague,
thank you so much for your interest to our manuscript and valuable comments! We agree with all of them. The manuscript has been significantly revised according to your recommendations. Please find our answers in the attached pdf-file.
Stay safe and healthy!
Best regards,
Roman Veselovskiy

Reviewer 3 Report
The well written manuscript presents new AFT data from mostly gabbroic rocks of the Archean Murmansk block. The authors found three age clusters, which are explained by different processes. C3 is explained by nearby Caledonian orogeny (reasonable for samples 3-4, but os obvious for the distant samples 11 and 12). C1 (180-190 Ma) is related to the Barents Sea magmatic province during large-scale plume-lithospheric interaction. The rather enigmatic C2-event (290-320 Ma) is explained by denudation and is virtually associated with remagnetization. Murmansk block. The authors found three age clusters, which are explained by different processes. C3 (ca. 422 Ma) is explained by nearby Caledonian orogeny (reasonable for samples 3-4, but not so obvious for the distant samples 11 and 12).
Specific remarks
- 27: Say why the track length distribution argues for a similar evolution.
- 42-43: “Our results roughly cover the "white spots" on the AFT map of NE Fennoscandia.”: Put this sentence earlier in the abstract.
- 55-57: Add a reference to: “Fission-track (FT) analysis is a sensitive low-temperature thermochronological method that is widely used to construct quantitative models of the thermal and tectonic evolution of various tectonic units.”
Figure 1: enlarge size of lettering of legend in Fig. 1.
Table 1: “Samsonov A.V., pers. comm.” (= co-author: better to write: A.V. Samsonov, unpublished
- 199: What is EDM?? Explain abbreviation in the method section.
- 214: better:” a restricted range of generally low U-contents”
- 253-255: Are these Cl contents in a Table?
- 259: Why not explain it as a distant effect of the alkaline Kola magmatic province?
- 279-280: Mention and discuss Fig. 5 (updated after Hendriks et al.. with updated low-temperature thermochronological database of NE Fennoscandia also at the end of the Discussion section, too.
Author Response

(The authors gave the same response as above.)
